# Mode conversion of hyperbolic phonon polaritons in van der Waals terraces

Byung-Il Noh [1,4], Sina Jafari Ghalekohneh[2,4], Mingyuan Chen [1], Jialiang Shen[1], Eli Janzen[3], Lang Zhou [1], Pengyu Chen[1], James H. Edgar [3], Bo Zhao [2] ✉ & Siyuan Dai [1] ✉

Electromagnetic hyperbolicity has driven key functionalities in nanophotonics, including super-resolution imaging, efficient energy control, and extreme light manipulation. Central to these advances are hyperbolic polaritons—nanometer-scale light-matter waves—spanning multiple energy-momentum dispersion orders with distinct mode profiles and incrementally high optical momenta. In this work, we report the mode conversion of hyperbolic polaritons across different dispersion orders by breaking the structure symmetry in engineered step-shape van der Waals (vdW) terraces. The mode conversion from the fundamental to high-order hyperbolic polaritons is imaged using scattering-type scanning near-field optical microscopy (s-SNOM) on both hexagonal boron nitride (hBN) and alpha-phase molybdenum trioxide ($\alpha$-MoO$_3$) vdW terraces. Our s-SNOM data, augmented with electromagnetics simulations, further demonstrate the alteration of polariton mode conversion by varying the step size of vdW terraces. The mode conversion reported here offers a practical approach toward integrating previously independent different-order hyperbolic polaritons with ultra-high momenta, paving the way for promising applications in nano-optical circuits, sensing, computation, information processing, and super-resolution imaging.

Electromagnetic hyperbolicity[1], characterized by opposite signs in the principal components of the permittivity tensor ($\varepsilon_i \varepsilon_j < 0$, where $i, j = x$, $y$ or $z$), drives important advances in nanophotonics. These include the exceptionally high photonic density of states, unconventional light propagation, and access to deep sub-diffraction optics. Such advances have led to valuable functionalities like super-resolution imaging[2–4], efficient energy control[5–8], and extreme light manipulation[9,10]. At the core of these advances and functionalities are high-momentum (high-$k$) hyperbolic polaritons—the nanometer-scale light-matter waves confined in hyperbolic materials. Following the Fabry-Pérot resonance condition[11], hyperbolic polaritons span multiple energy-momentum ($\omega$-$k$) dispersion branches (Fig. 1a, indexed by $l = 0, 1, 2, \ldots$) with incremental momenta $k$ and distinct mode profiles. These different-order hyperbolic polaritons can be studied using far-field Fourier Transform Infrared Spectroscopy on nano-patterned hyperbolic materials[12–14]. Real-space near-field studies typically image the lowest-$k$ zeroth-order ($l = 0$) hyperbolic polaritons as dominant single-period polariton fringes[11,15–23]. Two recent works on alpha-phase molybdenum trioxide ($\alpha$-MoO$_3$)[24,25] reveal high-order ($l > 0$) hyperbolic polaritons using geometric confinement at specific frequencies[24] and strong near-fields in nanowires[25], both of which directly launch high-order polaritons. In isotopically pure hBN[26], reduced phonon loss enables direct imaging of high-order polariton fringes. While different-order hyperbolic polaritons have been probed, their distinct momenta and mode profiles make them relatively independent, lacking evident interaction, integration, or conversion.

[1]Materials Research and Education Center, Department of Mechanical Engineering, Auburn University, Auburn, AL, USA. [2]Department of Mechanical Engineering, University of Houston, Houston, TX, USA. [3]Tim Taylor Department of Chemical Engineering, Kansas State University, Manhattan, KS, USA. [4]These authors contributed equally: Byung-Il Noh, Sina Jafari Ghalekohneh. ✉e-mail: bzhao8@uh.edu; sdai@auburn.edu

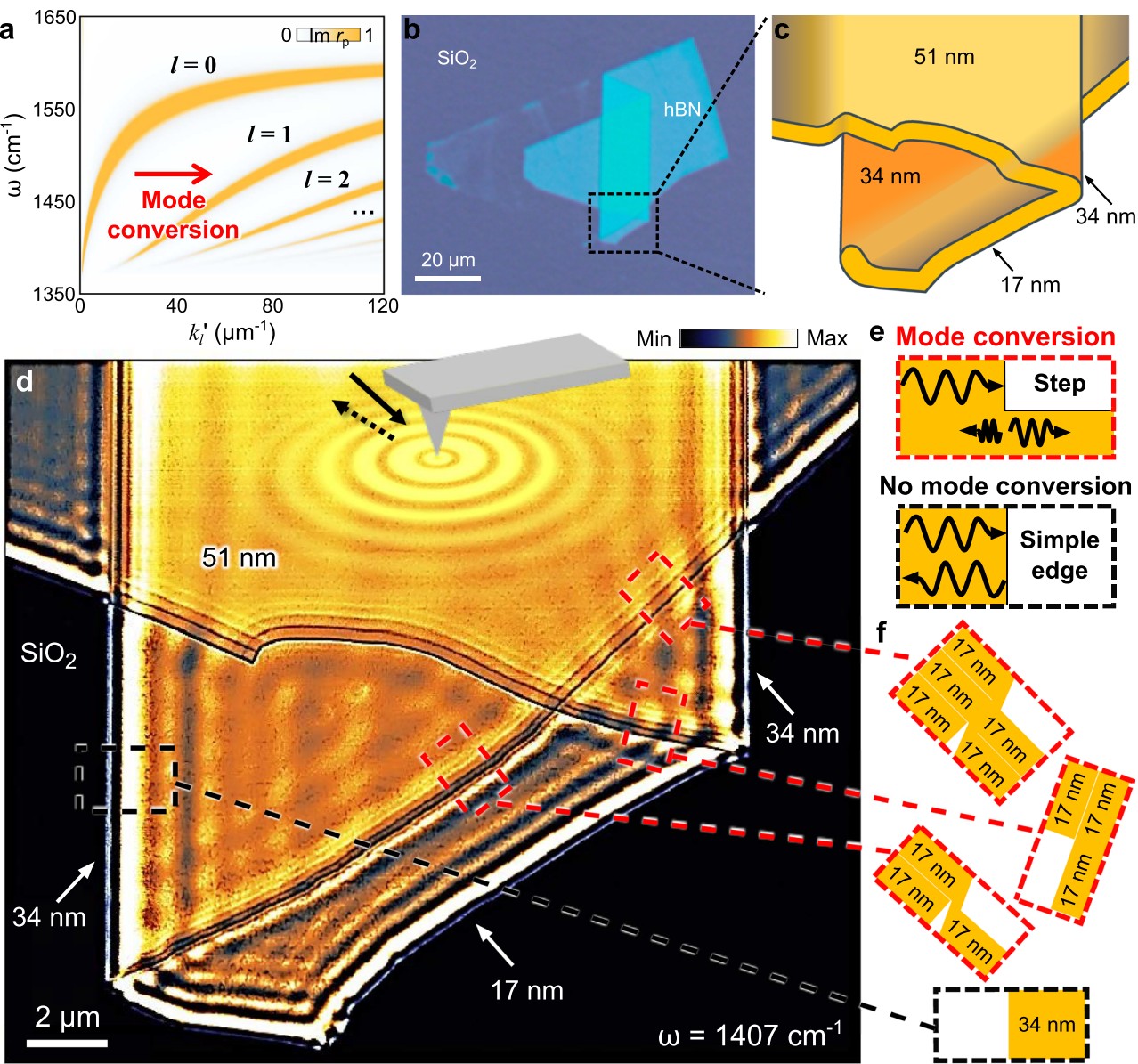

**Fig. 1 | Nano-infrared imaging of hyperbolic phonon polariton mode conversion in the hexagonal boron nitride (hBN) terrace using scattering-type scanning near-field optical microscopy (s-SNOM).** **a** The energy-momentum ($\omega$-$k$) dispersion of hyperbolic phonon polaritons in hBN (thickness: 34 nm) shown as a false-color map calculated by the imaginary part of the reflectivity, Im($r_p$). The red arrow indicates the mode conversion of the zeroth-order ($l = 0$) hyperbolic polariton into high-order ($l = 1, 2, ...$) hyperbolic polaritons. **b** The optical microscope image of terraced hBN made by the pickup-and-stack technique from a mechanically exfoliated hBN thin slab. **c** The schematic of the hBN terrace featuring a variety of simple slab, step, and covered step edges. **d** s-SNOM amplitude image of the terraced hBN at the infrared (IR) frequency $\omega = 1407$ cm$^{-1}$. The solid and dotted black arrows represent the incident and backscattered IR light at the s-SNOM tip (gray). The concentric orange circles delineate the hyperbolic phonon polariton waves launched by the s-SNOM tip in the terraced hBN. **e** Schematics of mode conversion and no mode conversion of hyperbolic phonon polaritons at the step-shape edge (top) and the simple slab edge (bottom), respectively. **f** Cross-section schematics of the step-shape edges (top) and simple slab edge (bottom) in (**d**).

In this work, we report mode conversions of different-order hyperbolic polaritons (e.g., arrow, Fig. 1a) by breaking structure symmetry in van der Waals (vdW) terraces with engineered step-shape edges. These mode conversions result from changes in polariton momentum $k$ upon reflection at asymmetric step-shape edges, in contrast to conventional symmetric simple slab edges that tend to preserve both momentum and mode profile during reflection. Moreover, the mode conversions of different-order hyperbolic polaritons can be altered by varying the step sizes at the asymmetric edges. Our combined experimental and theoretical results uncover the distinct nano-optical phenomenon of polariton mode conversions by simple vdW engineering. The mode conversion offers a practical approach towards integrating previously independent different-order hyperbolic polaritons with ultra-high momenta for advanced polariton nano-optical functionalities.

## Results

### Nano-imaging of polariton mode conversion in vdW terraces

The mode conversions between different-order hyperbolic polaritons were revealed through infrared nano-imaging of engineered vdW terraces using the scattering-type scanning near-field optical microscopy (s-SNOM, Fig. 1d). Hexagonal boron nitride (hBN) terraces were assembled using the pickup-and-stack technique[27] from mechanically exfoliated vdW thin slabs (Fig. 1b, c). It features both simple slab edges,

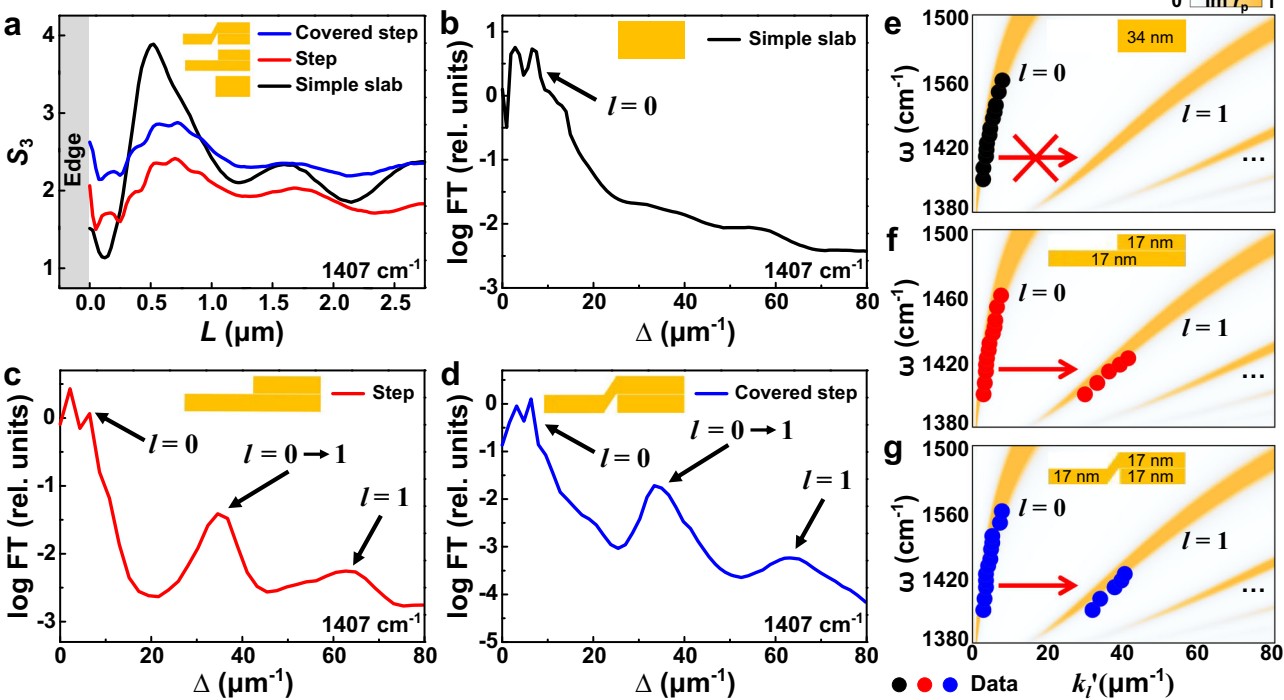

**Fig. 2 | s-SNOM line profiles and energy-momentum (ω-k) dispersions of hyperbolic phonon polaritons at various edges in the hBN terrace. a** s-SNOM line profiles taken from simple slab and step-shape edges in Fig. 1d. Fourier transform (FT) spectra of the s-SNOM line profiles from the symmetric simple slab edge (**b**), asymmetric step-shape edge (**c**), and asymmetric covered step-shape edge (**d**) in (**a**). In the simple slab edge (**b**), the FT peaks show zeroth-order hyperbolic phonon polaritons ($l = 0$) features from the standard edge-launch-photon-interference and tip-launch-edge-reflect-interference mechanisms. Unlike the symmetric simple slab edge, the asymmetric step-shape edges (**c, d**) reveal high-order hyperbolic phonon polaritons ($l = 1$) from the $l = 0 \rightarrow 1$ mode conversion. ω-k dispersions of hyperbolic phonon polaritons from the simple slab edge (**e**), step-shape edge (**f**), and covered step-shape edge (**g**), respectively. The thicker side thickness of the hBN is 34 nm. The s-SNOM data and simulation results are plotted with dots and false-color maps, respectively.

where the sample completely terminates (Fig. 1e, bottom), and step-shape edges, where the sample partially terminates on the top part, forming a step (Fig. 1e, top). The characterization tool s-SNOM is an illuminated atomic force microscope (AFM) that simultaneously records topography and nano-optical images of the underneath sample (Fig. 1d). The s-SNOM observable near-field amplitude $S_3$ (methods) with a spatial resolution of ~10 nm can map nano-optical phenomena in real space. On polaritonic materials, the s-SNOM tip acts as an antenna[28] to bridge the momentum mismatch and transfer energy between free-space light (wavelength $\lambda_f$ and frequency $\omega = 1/\lambda_f$) and polaritons[15,16].

In the hBN terraces, hyperbolic polaritons are imaged close to the simple slab edges and step-shape edges. At the simple slab edge (Fig. 1d, black dashed box), parallel fringes were observed, similar to previous s-SNOM works on polaritons[11,15–19]. They show the strongest oscillation closest to the edge, followed by damped ones away from the edge, as evidenced in the s-SNOM line profile as a function of the distance to the edge $L$ (Fig. 2a, black curve). Fourier Transform (FT) analysis (Fig. 2b) of the line profile from the simple slab edge reveals two evident resonances at $\Delta = 3$ and $6\,\mu m^{-1}$. They correspond to the standing wave interferences between the edge-launched polaritons and the free-space illumination and the interferences between the tip-launched and edge-reflected polaritons, respectively[29]. A systematic s-SNOM study at various ω reveals the ω-k polariton dispersion along the zeroth-order ($l = 0$) hyperbolic branch, as verified by our calculation in Fig. 2e. Briefly, hyperbolic polaritons span multiple ω-k dispersion branches[11]:

$$k = k'_l + ik''_l = -\frac{\Psi}{d}\left[\arctan\left(\frac{1}{\varepsilon_t \Psi}\right) + \arctan\left(\frac{\varepsilon_s}{\varepsilon_t \Psi}\right) + \pi l\right], \Psi = \frac{\sqrt{\varepsilon_z}}{i\sqrt{\varepsilon_t}}, \quad (1)$$

where $d$ is the hBN thickness, $l = 0, 1, 2, 3, \ldots$ is an integer and the branch index (mode order). $\varepsilon_s$ is the substrate permittivity, $\varepsilon_t = \varepsilon_x = \varepsilon_y$ and $\varepsilon_z$ are the in-plane and out-of-plane permittivities of hBN. $k = k'_l + ik''_l$ is the complex in-plane momentum of hyperbolic polaritons and relates to the polariton wavelength $\lambda_l$ by $k'_l = 2\pi/\lambda_l$.

While the data at hBN simple slab edges align with conventional s-SNOM studies[11,15–23] to probe hyperbolic polaritons mainly at the zeroth-order, our results at the step-shape edges (Fig. 1d, red dashed boxes) reveal distinct characteristics. In addition to the relatively long-period polariton fringes observed at simple slab edges, short-period beats superimposed on these fringes appear near a variety of step-shaped edges. These step-shaped edges include both regular and covered steps (Fig. 1f, red boxes). As shown in the s-SNOM line profiles (red and blue curves in Fig. 2a), these short-period beats decay from the step-shape edges into the sample interior, similar to the long-period fringes. The FT spectra in Fig. 2c, d reveal evident resonances $\Delta = 35\,\mu m^{-1}$ for these beats. They correspond to standing waves with a wavelength much smaller than the zeroth-order ($l = 0$) hyperbolic polaritons.

In hyperbolic materials, the observed short-period beats are indicative of high-order ($l > 0$) hyperbolic polaritons. However, due to their short propagation length and high $k$ (see Eq. 1)[11], standard tip-launch-edge-reflect and edge-launch-photon-interfere mechanisms do not produce evident interference for high-order hyperbolic polaritons, as evidenced by our data at the simple slab edge (Fig. 2b) and previous works[11,15–23]. Instead, the short-period beats are predominantly attributed to the mode conversion of hyperbolic polaritons from the zeroth-order branch to the high-order branches (arrow, Fig. 1a) when the former was launched by the tip, propagated, and reached the step-shape edges. Specifically, hyperbolic phonon polaritons propagate inside the vdW slab following a zigzag trajectory by reflecting at the

top and bottom surfaces. Upon reaching a simple slab edge (Fig. 1e, bottom), these polaritons reflect off the vertical sidewall without evident scattering, thereby preserving their original mode order and profile. In contrast, a step-shape edge (Fig. 1e, top) introduces a sharp step corner that can strongly scatter[22,30] the incoming polaritons inside the slab. This scattering provides additional momentum[22,30], enabling mode conversion by bridging the $k$-mismatch between polaritons of different orders. As a result, zeroth-order ($l = 0$) polaritons are converted into first-order ($l = 1$) modes upon reflection at the step-shaped edge. In our experiment, these converted $l = 0 \rightarrow 1$ first-order polaritons propagate back towards the s-SNOM tip and interfere with the newly launched $l = 0$ polaritons, forming standing wave interferences between the tip and the step-shape edge (see detailed analysis of the interference mechanism in Supplementary Note 1). As the sample is scanned underneath the tip, the standing wave interference is recorded as short-period beats in our s-SNOM image (Fig. 1d). Notably, high-order hyperbolic polaritons exhibit much shorter propagation lengths than their zeroth-order counterpart. Therefore, without the mode conversion, it is difficult for high-order polaritons to complete a tip-launch-edge-reflect round trip to produce evident standing wave interferences at either simple slab edges (Fig. 1d black dashed box and ref.[11,15–23]) or step-shape edges. Note that weak signals around $65 \, \mu m^{-1}$—by its value, may correspond to the tip-launch-edge-reflect $l = 1$ polariton fringes—cannot correlate with dominant s-SNOM signatures, especially the $l = 0 \rightarrow 1$ beats (see the FT decomposition of our s-SNOM data in Supplementary Note 2).

The observed mode conversion is corroborated in the ω-$k$ dispersion of hyperbolic polaritons (Fig. 2f-g). The standing wave interference between the converted first-order and the newly launched zeroth-order polaritons produces a periodic resonance of $\Delta = k_0' + k_1'$ (Supplementary Note 1). Therefore, the momentum $k_1'$ of the converted $l = 0 \rightarrow 1$ hyperbolic polaritons can be extracted from the FT spectra in Fig. 2c, d. These extracted s-SNOM data (red and blue dots) from both the step-shape edge and the covered step-shape edge reveal a systematic ω dependence and agree well with our ω-$k$ dispersion calculation (false color maps) in Fig. 2f, g, thereby validating the polariton mode conversion mechanism described above.

Polariton mode conversion induced by asymmetric step-shape edges is expected to be generic to hyperbolic materials featuring multiple ω-$k$ dispersion branches. In Fig. 3, we studied this effect in terraces made of another representative hyperbolic vdW material, α-MoO$_3$[31–33]. The α-MoO$_3$ terraces were fabricated using the similar pickup-and-stack technique[27] as for hBN (Fig. 1). The s-SNOM amplitude images of a simple slab α-MoO$_3$ (Fig. 3a) and a terraced α-MoO$_3$ with a step-shape edge (Fig. 3b) exhibit distinct features. Consistent with the results in hBN (Figs. 1, 2), long-period polariton fringes are observed near the edge of the simple slab (Fig. 3a). In contrast, near the step-shaped edge (Fig. 3b), short-period beats are superimposed on these fringes, indicating the polariton mode conversion. FT analysis of the s-SNOM line profiles (Fig. 3c, d) reveals clear signatures of the $l = 0 \rightarrow 1$ mode conversion at the step-shaped edge (Fig. 3f). The extracted momentum $k_1'$ (red dots, Fig. 3h) shows excellent agreement with the ω-$k$ dispersion calculation (false color map, Fig. 3h).

### Numerical simulation to verify the polariton mode conversion
In order to further verify the polariton mode conversion phenomenon, we carried out total-field scattered-field electromagnetic simulations using the Finite Difference Frequency Domain (FDFD) method (see details in Supplementary Note 3). We calculated the electric field $E_x$ at the cross-sections of the simple slab and step-shape hBN terraces (Fig. 4e, g). Hyperbolic polaritons are launched from the black dashed lines and propagate along the $+x$ direction. Different orders of hyperbolic polaritons show distinct mode profiles (Fig. 4a–d) following their Fabry-Pérot resonances indexed by $l$[11,34]. They possess different symmetries and are either even ($l = 0, 2, \ldots$) or odd ($l = 1, 3, \ldots$) to

the hBN slab centerline. Upon reaching the edges, these polaritons get reflected, propagate along the $-x$ direction, and are finally analyzed at the left side of the source, where the polariton reflectivity at various branches can be extracted using mode orthogonality. These FDFD simulations support our experimental results by revealing that edge symmetry is crucial in polariton mode conversion. During the reflection, while the symmetric simple slab edge preserves polaritons in the same order (same $l$), the asymmetric step-shape edge does not, leading to strong mode conversion. For example, in hBN structures, input from our experiments in Fig. 1, zero-order ($l = 0$) hyperbolic polaritons are launched before the simple slab and step-shaped edges. The FT results reveal that the simple slab edge mainly reflects these polaritons into the zeroth order (Fig. 4f). In contrast, the step-shape edge reflects them into both the zeroth and the first order (Fig. 4h), indicating an evident $l = 0 \rightarrow 1$ polariton mode conversion. In Fig. 4i, j, we quantify the mode conversion rate using reflectivity $R_{ij}$—the power ratio between the reflected $j$-th order mode due to the incident $i$-th order mode—for more generalized cases where the incident hyperbolic polaritons are at $l = 0, 1, 2$, and $3$. Notably, the symmetric simple slab edge preserves the polariton modal symmetry (Fig. 4i): $R_{00}$, $R_{11}$, $R_{22}$, and $R_{33}$ are significantly larger than others, indicating that even (odd) modes $l = 0, 2, \ldots$ (1, 3, …) are only reflected into even (odd) modes. Conversely, the asymmetric step-shape edge reflects polaritons into both odd and even modes, regardless of the order of the incident polaritons (Fig. 4j) —demonstrating that breaking edge symmetry leads to polariton mode conversion.

### Altering the polariton mode conversion in vdW terraces
After demonstrating the mode conversion of hyperbolic polaritons at asymmetric step-shape edges, we investigate the alteration of mode conversion by varying the step size. In Fig. 5, we study the polariton mode conversion in hBN terraces where all thicker sides of their step-shape edges have identical thicknesses $H = 51$ nm, but thinner sides have different thicknesses $h$. Figure 5b, c present representative s-SNOM images of these hBN terraces with thinner side thicknesses $h = 17$ and 34 nm ($h/H = 0.33$ and $0.67$). The clear differences in the imaged polariton fringes and beats in Fig. 5b, c suggest a strong dependence of mode conversion on the step size. This trend is further revealed by the s-SNOM line profiles (Fig. 5d) and the extracted $l = 0 \rightarrow 1$ polariton line profiles from the inverse FT analysis (Fig. 5e, more details about the FT analysis are provided in Supplementary Note 4). Among all the terraces studied, the one with $h/H = 0.33$ exhibits the strongest $l = 0 \rightarrow 1$ oscillation (red profile). Since all incident zeroth-order (tip-launched) hyperbolic polaritons are identical (Fig. 5a, all $H = 51$ nm), the terrace with a step-shape edge of $h/H = 0.33$ yields the highest $l = 0 \rightarrow 1$ mode conversion rate $R_{01}$. For both smaller and larger $h/H$, the converted first-order polariton oscillations are weaker, corresponding to lower $l = 0 \rightarrow 1$ mode conversion rates $R_{01}$. The mode conversion rate $R_{01}$ can be quantified by measuring the oscillation amplitude of the converted $l = 0 \rightarrow 1$ polariton line profiles[35,36]. In Fig. 5f, we plot the dependence of the mode conversion rate on the step ratio $h/H$ (see details in Supplementary Note 5), where the experimental data from the s-SNOM imaging (squares) agree excellently with the FDFD simulation results (curve).

### Discussion
Combined s-SNOM nano-imaging data, electromagnetics modeling, and FDFD simulations in Figs. 1–5 demonstrate the mode conversion of hyperbolic polaritons in vdW terraces. The mode conversion originates from the strong scattering at asymmetric step-shape edges that bridges the momentum mismatch and transfers the energy between hyperbolic polaritons across different-order dispersion branches. The observed different-order polariton mode conversion can be altered by varying the step size in the terraced structures. Therefore, the mode conversion demonstrated in our work provides a practical approach

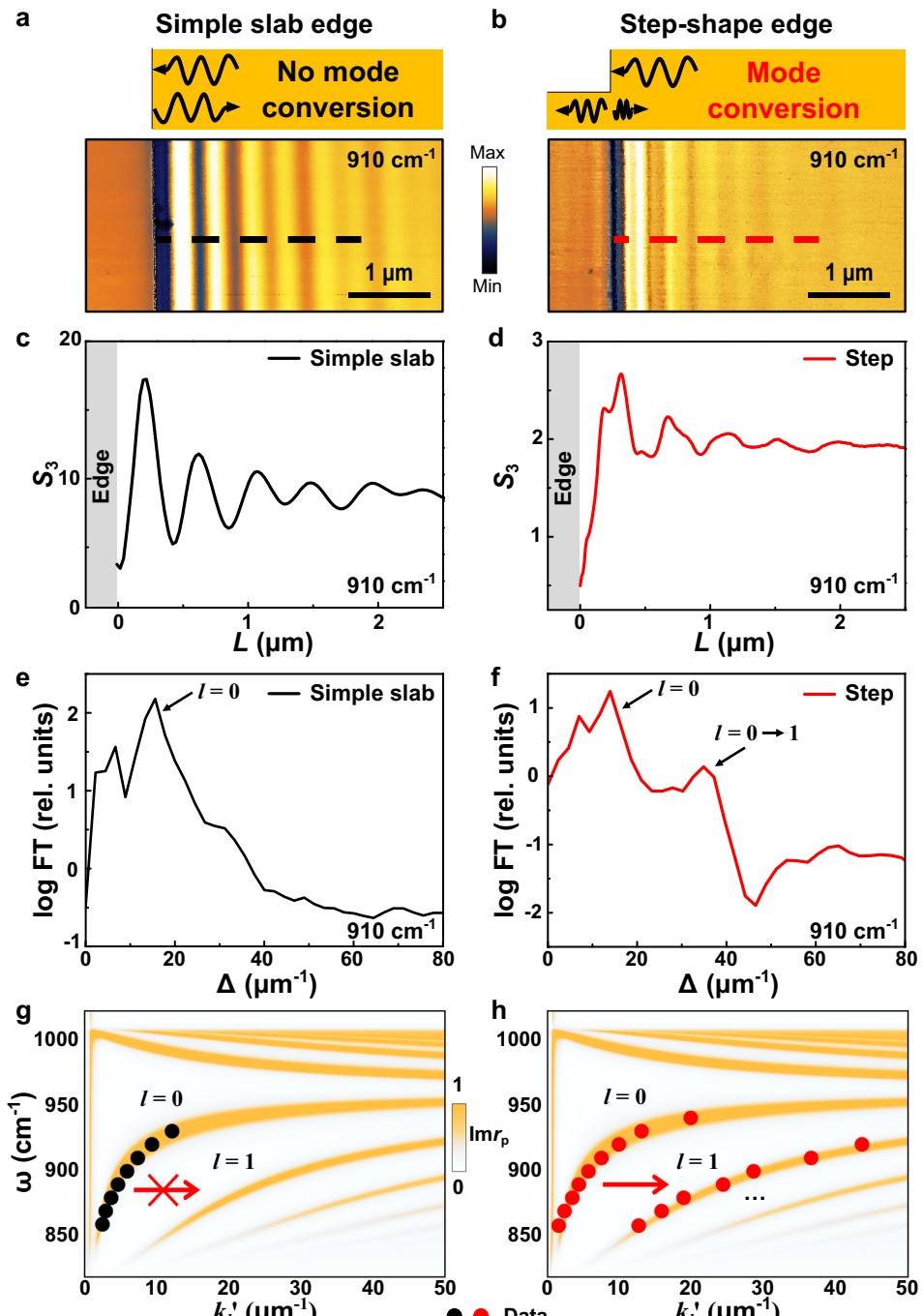

**Fig. 3 | Mode conversion of hyperbolic phonon polaritons in alpha-phase molybdenum trioxide ($\alpha$-MoO₃) terraces.** Schematics and s-SNOM images of polaritons around the $\alpha$-MoO₃ simple slab edge (thickness: 160 nm) (**a**) and step-shaped edge (left and right thicknesses = 81 and 165 nm, respectively) (**b**) at $\omega = 910$ cm⁻¹. Unlike the symmetric simple slab edge (**a**), the asymmetric step-shape edge (**b**) of the $\alpha$-MoO₃ terrace reveals beats indicating a similar hyperbolic polariton mode conversion to those in hBN terraces (Figs. 1, 2). **c**, **d** s-SNOM line profiles extracted from the simple slab (black dashed line) and step-shape (red dashed line) edges in (**a**, **b**). **e**, **f** FT spectra of the s-SNOM line profiles from the symmetric simple slab edge and asymmetric step-shape edge in (**c**, **d**). $\omega$-$k$ dispersion of hyperbolic phonon polaritons of $\alpha$-MoO₃ from the simple slab edge (**g**) and the step-shape edge (**h**). The s-SNOM data and simulation results are plotted with color dots and false-color maps, respectively.

toward integrating previously independent different-order hyperbolic polaritons with ultra-high momentum and precious figures of merit for advanced polariton nano-optical functionalities. Future works may be directed toward tailoring edges or implementing metastructures at the edges for on-demand polariton mode conversion and propagation redirection[37] at specific orders. The polariton mode conversion may be varied by delicately controlling the steepness of the asymmetric edges

(Supplementary Note 6). In addition, polariton launchers may be implemented on one side of the vdW terrace to study the polariton transmission and mode conversion across the step-shape edges. Moreover, it is worth integrating the mode conversion structures into practical nano-optical devices for nano-optical circuits, sensing, computation, energy transfer, information processing, and super-resolution imaging.

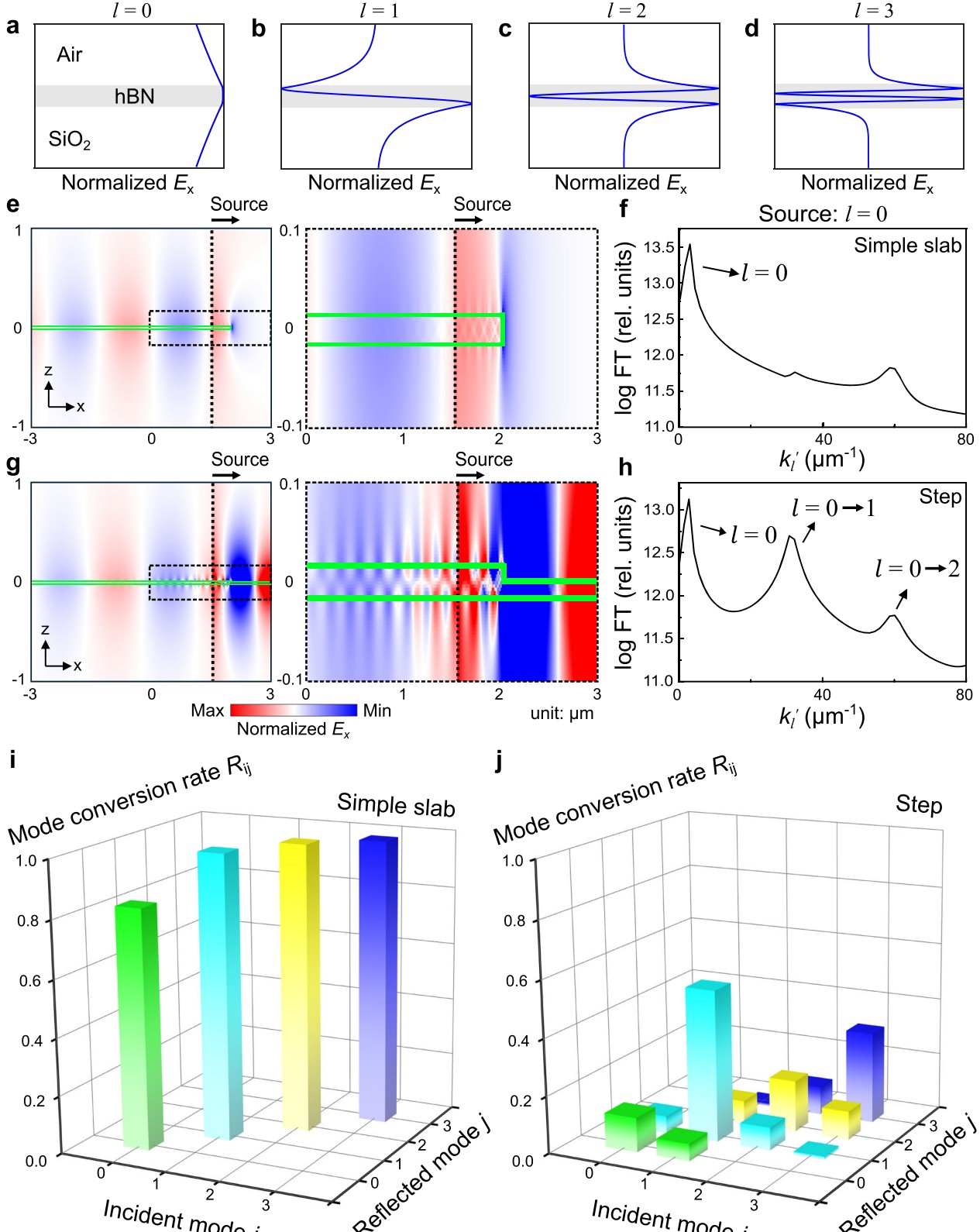

**Fig. 4 | Finite-difference frequency-domain (FDFD) simulation of hyperbolic polariton mode conversions in the hBN terrace.** The normalized $x$-component of electric field ($E_x$) for (**a**) zeroth-order, (**b**) first-order, (**c**) second-order, and (**d**) third-order hyperbolic phonon polaritons. False color maps of $E_x$ at the cross-sections of the simple slab (symmetric, **e**) and step-shape (asymmetric, **g**) hBN terraces, with the incidence of zeroth-order hyperbolic polaritons at the black dashed lines. The green boxes denote the locations of the simple slab hBN and the step-shaped hBN terraces in our simulations. In the simple slab edge (**e**), the left side is extended to infinity. In the step-shape edge (**g**), both sides are extended infinitely. **f, h** FT spectra of the reflected fields in (**e**) and (**g**), respectively. The momenta $k_l'$ of various-order hyperbolic polaritons agree well with our experimental results in Fig. 2. The arrows mark the polariton modes at different branches. **i, j** 3D bar graphs of reflectivity $R_{ij}$: the energy ratio between the reflected $j$-th order mode due to the incident $i$-th order mode from hBN terraces featuring the simple slab and step-shape edges, respectively.

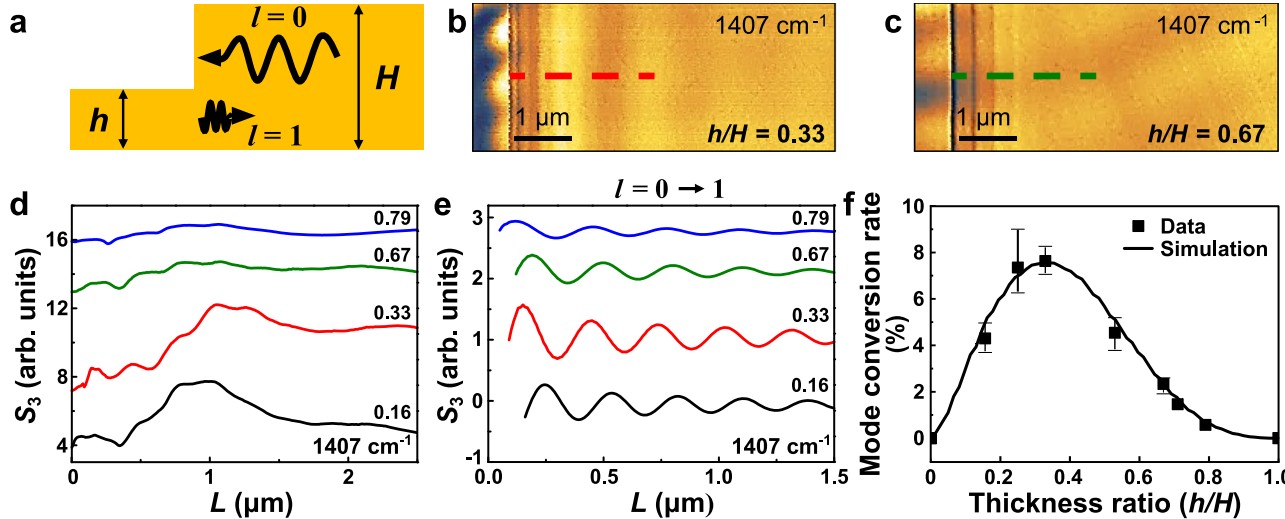

**Fig. 5 | Altering hyperbolic polariton mode conversion by varying the step size.** **a** The schematic of the hBN terrace featuring a step-shape edge. $h$ and $H$ indicate the thinner and thicker sides of the hBN, respectively. **b, c** Representative s-SNOM images of step-shape edges with thinner side thicknesses $h = 17$ and $34$ nm ($h/H = 0.33$ and $0.67$). s-SNOM line profiles (**c**) are obtained from red and dark green dashed lines. **d** s-SNOM line profiles over step-shaped edges with $h/H = 0.16$, $0.33$, $0.67$, and $0.79$. **e** Converted $l = 0 \rightarrow 1$ polariton profiles taken from the inverse FT of the s-SNOM line profiles in (**d**). **f** The dependence of the mode conversion rate $R_{01}$ on the step ratio $h/H$. The thickness of the thicker side is $H = 51$ nm. Error bars represent the observed range (minimum–maximum) of mode conversion rates.

During the manuscript peer review process, we became aware of a related study reporting boundary-excited high-order hyperbolic phonon polaritons and the pseudo-birefringence effect in α-MoO₃[38].

## Methods
### Fabrication of vdW terraces
The hBN and α-MoO₃ terraces featuring simple slab and step-shape edges were assembled using the standard vdW dry transfer method. hBN and α-MoO₃ bulk crystals were grown using Cr–Fe flux with a temperature gradient[39] and thermal physical vapor deposition[40]. The vdW slabs were mechanically exfoliated from bulk crystals and transferred onto Si wafers with 300 nm thick thermal oxide. Using a poly (bisphenol A carbonate)/polydimethylsiloxane stamp, vdW terraces were assembled by partially picking up the exfoliated large vdW slab and stacking that part on top of the original slab.

### Infrared nano-imaging
The infrared nano-imaging of hyperbolic polariton mode conversion in the terraced vdW was conducted using the scattering-type scanning near-field optical microscope (s-SNOM, www.neaspec.com). The s-SNOM is based on a tapping-mode AFM. The PtIr-coated AFM tip with a radius of ~10 nm (Arrow-NCPt, NanoWorld AG, Switzerland) was illuminated by monochromatic Mid-IR quantum cascade lasers (QCLs, www.daylightsolutions.com) with frequency spanning 845–1800 cm⁻¹. The nano-IR s-SNOM images were recorded by a pseudoheterodyne interferometric detection module with a tapping frequency of 245–280 kHz and tapping amplitude of ~70 nm. The detected optical signal was demodulated at the third harmonics ($S_3$) of the tapping frequency in order to obtain the pure near-field signal.

## Data availability
Relevant data supporting the key findings of this study are available within the article and the Supplementary Information file. All raw data generated during the current study are available from the corresponding authors upon request.

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

## Acknowledgements

S.D. acknowledges the support from the National Science Foundation under Grant Nos. DMR-2238691 and DMR-2525882, and ACS PRF fund 66229-DNI6. B.-I.N. and J.S. acknowledge financial support from the Alabama Graduate Research Scholars Program (GRSP), funded through the Alabama Commission for Higher Education and administered by the Alabama EPSCoR. B.Z. and S.J.G. acknowledge the funding from the University of Houston through the SEED program and the National Science Foundation under Grant No. CBET-2314210, and the support of the Research Computing Data Core at the University of Houston for assistance with the calculations carried out in this work. Support for hBN crystal growth was provided by the Office of Naval Research, award number N00014-22-1-2582. S.J.G. and B.Z. thank Drs. Jiahui Wang and Nathan Zhao for the discussions on the FDFD calculations.

## Author contributions

S.D. conceived the idea. B.-I.N., M.C., and J.S. fabricated the device and performed the optical experiments. L.Z. assisted with the plasma cleaning of the substrates for the van der Waals terraces. E.J. and J.E. provided the hBN crystals. S.J.G. and B.Z. conducted the theory and simulations. B.-I.N., S.D., S.J.G., and B.Z. analyzed the data. S.D., B.Z., J.E., and P.C. supervised the project.

## Competing interests

The authors declare no competing interests.
