## [Transparent Peer Review file · Nature Communications]

Mode Conversion of Hyperbolic Phonon Polaritons in van der Waals terraces

Corresponding Author: Professor Siyuan Dai

Version 0:

Reviewer comments:

Reviewer #1

(Remarks to the Author)

The manuscript "Mode Conversion of Hyperbolic Phonon Polaritons in van der Waals terraces" presents a study in which step-shaped van der Waals (vdW) terraces are used to launch higher-order hyperbolic phonon polaritons in two representative hyperbolic materials, hBN and MoO₃, and refers to this phenomenon as mode conversion from the fundamental to higher-order modes. I like the idea and find the work interesting, especially in the way it introduces the concept of mode conversion to the phonon-polariton community.

However, my biggest concern is the lack of quantitative information on the conversion the authors emphasize. When discussing conversion, efficiency is a key metric, yet no absolute conversion efficiency is reported until the last figure, and even there only a normalized conversion rate is provided. What is the absolute conversion efficiency measured in such configuration, and how does it compare with the simulations (energy conversion)? Without absolute conversion efficiency it is difficult to assess practicality or to benchmark against existing methods.

In addition, how does this terrace-based method of launching higher-order modes compare with other approaches, such as using isotopically pure materials as the authors have previously demonstrated (<https://doi.org/10.1038/nmat5047>) or using nanowires (Ref. 25 in the main text)?

I also have the following minor comments:

1. It is suggested to visualize the data in Figure 4i-j graphically instead of list all values in a table.
2. More detail needs to be added on how the data were processed from Figure 5d to Figure 5e. Although the reviewer understood the process, a clearer explanation would make it accessible to a broader readership of Nature Communications.
3. A relevant work could be cited and compared (<https://doi.org/10.1021/acsnano.4c03630>).

Reviewer #2

(Remarks to the Author)

The authors report mode conversion between different dispersion orders of hyperbolic polaritons achieved through a stepped van der Waals (vdW) terraces. Higher-order modes of hBN and α -MoO₃ were imaged using s-SNOM, respectively. However, I think the novelty of this proposition may not be sufficient to support its publication in Nature Communications. The excitation at edges and the observation of higher-order hyperbolic polaritons have already been extensively investigated in this field, as exemplified by the previous reports on hBN and its isotopic heterostructures, as well as the higher-order hyperbolic polaritons in α -MoO₃ launched by 3C-SiC nanowires. Thus, the authors' research represents only a tiny part of the polariton field, and its significance is relatively limited.

Additionally, the manuscript is not organized well, which makes it difficult for readers to understand.

1. Figure 1 appears somewhat disorganized, with the numbering positions seeming inappropriate. Furthermore, the representation of the red dashed boxes is unclear—should an additional schematic be included to illustrate the relative

positions between layers?

2. In the third line of the third paragraph on page 3, the authors mention "Hexagonal boron nitride (hBN) and α -MoO₃ terraces were assembled using the tear-and-stack technique from mechanically exfoliated vdW thin slabs," which led me to mistakenly assume that Figure 1 contained a heterostructure of hBN and α -MoO₃. However, neither Figure 1 nor Figure 2 seems to include any content related to α -MoO₃. Is this worth mentioning?

3. In Figure 2, panels (c) and (d) refer to "step" and "covered step" structures, respectively, but there is no comparative discussion distinguishing the two. Can it be assumed that there is no difference between them?

4. In Figure 4, the caption does not explain what the green solid boxes represent—should this be clarified? Additionally, in Figures (f) and (h), there appears to be a distinct peak at $l = 0 \rightarrow 2$. Why is this the case?

Reviewer #3

(Remarks to the Author)

Noh et al. report on near-field imaging results of hyperbolic polaritons from terraced flakes of hBN and claim mode conversion between the fundamental $l = 0$ hyperbolic thin film mode into the $l = 1$ mode upon reflection at a stepped etch. The data are beautiful and convincing, however, I have some fundamental doubts about the claim of "mode conversion" as detailed below. Therefore, I cannot recommend publication of this work in Nature Communication in its current form.

Detailed comments:

1) Unless I am missing something, I see a fundamental issue in the interpretation of the emergence of higher-order polariton fringes as originating from mode conversion. Let me explain my concern:

Throughout the paper, the authors assume that the tip launches a fundamental $l=0$ mode, that is then converted into a $l=1$ mode upon reflection at the stepped edge. I don't quite understand how this assumption is justified. Previous work, e.g. Giles et al. Nat. Mat. 17, 134 (2018), which the authors fail to cite, has observed higher order modes without stepped edges, meaning that, in principle, higher order mode launching and detection is possible in hBN, but is usually suppressed from materials losses. What the current work successfully shows is that, in addition, the reflection of odd-order modes at simple edges is suppressed (not 0 as above work shows), but enabled again at stepped edges.

I see the data in the following light: indeed, the stepped edges enable reasonably efficient reflection of odd-order hyperbolic modes, enabling their observation in a tip-launched edge-reflected scenario. However, this does not mean that the $l=1$ modes emerge from mode conversion, but simply that the odd higher-order modes launched at the tip are made visible by the stepped edge. I believe this is a reflection phenomenon rather than a conversion.

2) The problem of interpretation/assumption is particularly apparent in figure 4 where the authors explicitly assume an $l = 0$ fundamental mode as source, excluding the $l = 1$ component contained in the tip-launched hyperbolic mode. Without a scale on the log-plots (f,h), it is also difficult to gauge how efficient the mode conversion process is.

Furthermore, it is very difficult to grasp in Fig. 4 where the slab edge (simple or stepped) are in these plots.

3) I am struggling with the momentum conservation concept for the mode conversion. Since the momentum of the $l=1$ mode is not a harmonic multiple (typically) of the $l = 0$ mode, I am not actually sure how a coherent mode conversion would work. For a harmonic (in frequency domain commonly known as high harmonic generation), it would be more obvious, but I would expect that a non-integer harmonic would in fact largely cancel out. So I don't quite understand this conceptually, and would urge the authors to elaborate.

4) In the conclusion, the authors claim not just mode conversion but also energy redistribution. This is a bit odd since all modes are at the same energy.

Version 1:

Reviewer comments:

Reviewer #1

(Remarks to the Author)

The authors have addressed most of the reviewers' comments. With the new revised version, I have a few minor comments.

1. The new Figure 1 is still busy and crowded. Maybe it could be split into two figures.

2. The reported absolute conversion efficiency in SI is very low (below 8%). Can the authors comment on this? Also, I still think the absolute value should be reported in the main text instead of the normalized value currently shown there.

3. Although the authors confirm that the higher-order mode arises from mode conversion, could a boundary-induced excitation of the higher-order mode also play a role here, or could such a mechanism explain the observed phenomenon, as demonstrated in a recent work (<https://doi.org/10.1038/s41566-025-01755-5>)?

Reviewer #2

(Remarks to the Author)

1. The claimed novelty of mode conversion is unconvincing and insufficiently supported by evidence.

The central claim of the mode conversion mechanism at asymmetric step edges is not adequately distinguished from prior reports in this field. The manuscript fails to articulate a clear conceptual advance beyond the established framework. Crucially, the experimental data and analysis presented are insufficient to support this claim robustly. Furthermore, the treatment of high-order polaritons remains superficial, focusing on their appearance rather than a deep exploration of their converted modal properties, efficiency, or further tuning. The authors are encouraged to refer to recent in-depth studies on higher-order polaritons (DOI: 10.1002/adma.202300301; DOI:10.1038/s41566-025-01755-5) to see the level of mechanistic insight and evidence required for a high-impact claim.

2. Poor organization and confusing figure presentation undermine the clarity.

The figures, which are critical for understanding the nano-optical phenomena described, are poorly organized and inconsistently labeled. High-quality visual presentation is a fundamental aspect of scientific communication, and the current standard falls short of what is expected for a journal of this caliber.

Therefore, due to the unsubstantiated novelty of the proposed mode conversion mechanism and the poor clarity in figure presentation that significantly hinders comprehension, I find that the manuscript does not meet the high standards expected for publication in Nature Communications.

Reviewer #3

(Remarks to the Author)

Noh et al. responded well to my previous review as well as the other reviews. Conceptually, I see now how one would distinguish the edge-converted from the directly launched higher order modes. However, apart from the mathematical explanation in section S1, I do not see this proven in any of the data. What I would need to see is where in the FT spectrum to expect the direct (tip-launched edge-reflected) higher-order modes vs. edge-converted. It is in fact quite misleading, e.g., in Fig. 2 to show the dispersion (where all this treatment/extraction was already done) is plotted right next to the experimental data on different scales, absolute and relative, respectively, so by eye I would think (and was thinking that in the first round of reviews) that the data and the dispersion scales are the same (apart from a factor of 2). The $l=0$ momentum is so small, that the ratio between $l=0$ and $l=1$ momenta is qualitatively similar to the FT peak positions for $l=0$ and $l=0 \rightarrow 1$. If I understand correctly (now), the $l=1$ direct launching peak would emerge at almost twice the momentum as the $l=0 \rightarrow 1$ peak?

This is a critical point that needs to be made abundantly clear, and, honestly, it is not clear at all, not in any of the main text figures, nor in the SI. Part of the issue still is the simulations explicitly only exciting the $l=0$ mode. Or, like in Fig. S3, this critical comparison is not done. If there, I would compare the $l=1$ peak position with $l=1$ source, and the $l=0 \rightarrow 1$ peak position with $l=0$ source, it would be clear - bam. But the author never make this comparison visually or in numbers.

In fact, it is worse: in Fig 3 e,f and probably also in Fig. 2 c,d, the range of the FT spectra are cropped such that the any directly imaged $l=1$ reflected mode would not even be in the range of the plot. So, the data proving this mechanism are not actually shown!

I understand the experiment better now, and also how to distinguish the different schemes. But the manuscript still falls short on demonstrating this key part of the work. After all, this is what the story is about.

One additional comment: a much cleaner way of demonstrating mode conversion was just published in Nature Photonics (<https://doi.org/10.1038/s41566-025-01755-5>), and the authors must at least mention this work (The authors became aware of a competing work that appeared during preparation of this manuscript ... or something like that). Importantly, this other work uses nano-antennas instead of the tip to launch the polaritons, making much clearer images and a much cleaner analysis. Additionally, they also explicitly proof that the momentum parallel to the edge is conserved during the conversion, as to my previous question about momentum conservation.

Overall, I now better understand the work and don't question the validity of the claim. Yet, the authors should significantly revise the manuscript, to really convince the reader (with data!) of the proposed conversion indeed being proven.

Version 2:

Reviewer comments:

Reviewer #3

(Remarks to the Author)

Noh and coworkers have improved some of the aspects that I (and the other reviewers) had raised. I am still not particularly happy with the way the story is presented, brushing over the key observations that distinguish the current work from many others in the field. For instance, Eq. 1, has been shown in many previous papers. Instead, I would appreciate an Eq. that specifically discussed the mode conversion. However, I feel that with minimal changes during each iteration of the review process, this is not going to change much.

I do have some concerns, however, specifically relating to the changes included in the last iteration. As per my request, the

authors now show the full momentum range including the $l=1$ tip-launched edge reflected resonance in Fig. 2c,d that I had wondered about. In the SI, they also discuss this feature as " $l=1$ tip". However, in the revised main text, the authors state "Note that weak signals around $65 \mu\text{m}^{-1}$ cannot correlate with observable s-SNOM signatures", and support this statement with SI Fig. S2c. I find this whole argument (I cannot see the difference by eye if I remove the respective Fourier component) highly problematic and - in fact wrong. The Fourier component for the $l=1$ tip contribution is clearly observed in the data, and only because I don't "see it" in the real space traces doesn't make it less part of the data. Even more so, in the main text and Fig. 2c,d, the authors do not even call this by its name, apparently trying to hide the fact that there is a $l=1$ tip signal. This is borderline in terms of good scientific practice. I strongly advise against such a practice. More importantly, I don't understand why the authors do this. There is enough prior work that shows that higher order modes can indeed be observed also without mode conversion, as for instance pointed out by reviewer 2. From this current work, it is clear that the efficiency of mode conversion to $l=1$ is much higher than the direct excitation/edge reflection. To be more quantitative, the authors could in fact use these data to prove the claim, since the mode-converted $l=1$ contribution should have a different decay length than the $l=1$ tip contribution.

Either way, I see that there is significant resistance from the authors to tell a clearer story (I agree with reviewer 2 in that regard, even though I tried to be more specific in my reviews as to what is unclear). I leave it up to the editor to decide how to proceed but don't see the point of yet another round of reviews.

One more point: I find it disconcerting to quote only parts of the reviewers' statements in the response letter, ripping them out of context in several cases. I do not approve of this procedure. It is still up to the authors how they respond to these comments (or which parts thereof), obviously, but to remove part of the reviewers' comments distorts the view in an inappropriate way in my view.

Response to referees' comments

Manuscript #: NCOMMS-25-54301-T

Responses to Reviewer 1

We thank the reviewer for finding our results “*I like the idea and find the work interesting, especially in the way it introduces the concept of mode conversion to the phonon-polariton community.*” In the following, we respond to the comments of this reviewer.

Comment 1-1: *However, my biggest concern is the lack of quantitative information on the conversion the authors emphasize. When discussing conversion, efficiency is a key metric, yet no absolute conversion efficiency is reported until the last figure, and even there only a normalized conversion rate is provided. What is the absolute conversion efficiency measured in such configuration, and how does it compare with the simulations (energy conversion)? Without absolute conversion efficiency it is difficult to assess practicality or to benchmark against existing methods.*

Response to 1-1: We thank the reviewer for mentioning the absolute mode conversion rate. Following the reviewer’s suggestion, we have added Supplementary Section S4 in the revised manuscript to describe the processes of extracting the mode conversion rate. Specifically, we have plotted the combined experimental data and simulation results for the absolute mode conversion rate in Figure S5b. Below, we copy this added section.

S4. The extraction of $l = 0 \rightarrow 1$ polariton mode conversion rate R_{01}

The $l = 0 \rightarrow 1$ polariton mode conversion rate R_{01} can be extracted by measuring the intensity A of the polariton fringe oscillations. This oscillation intensity A can be quantified by fitting the $l = 0 \rightarrow 1$ s-SNOM line profiles (Figure 5e) with the envelope of a sinusoidal wave function Ae^{kL} .^{1,2} For example, the $l = 0 \rightarrow 1$ polariton line profile from the hBN terrace with a step ratio $h/H = 0.33$ can be fitted with $A = 0.26$ (Figure S5a). The mode conversion rate R_{01} can be obtained by normalizing the A at each h/H to that of the reflected $l = 0$ mode at the simple slab edge, which possesses most energy of the incident $l = 0$ mode. The absolute R_{01} can be extracted (Figure S5b) by leveraging the simulated $R_{00} = 0.71$. The relative mode conversion rate in Figure 5f of the main text was obtained by normalizing the data to the maximum in Figure S5b.

Figure S5 | Polariton mode conversion rate R_{01} . **a**, The extraction of the oscillation intensity of the $l = 0 \rightarrow 1$ polaritons by fitting the s-SNOM line profile with a sinusoidal wave function Ae^{kL} , at a representative step ratio $h/H = 0.33$. **b**, The dependence of the absolute mode conversion rate R_{01} on the step ratio h/H . The thickness of the higher side is $H = 51$ nm.

Comment 1-2: *In addition, how does this terrace-based method of launching higher-order modes compare with other approaches, such as using isotopically pure materials as the authors have previously demonstrated (<https://doi.org/10.1038/nmat5047>) or using nanowires (Ref. 25 in the main text)?*

Response to 1-2: We thank the reviewer for mentioning the two related studies. The novelty of our work lies not in directly “*launching high-order modes*”, but rather in demonstrating mode conversions between different hyperbolic polariton branches. Specifically, we show that asymmetric edges enable mode conversion between polaritons of distinct energy–momentum (ω - k) branches, as highlighted by the “ $l = 0 \rightarrow 1$ ” conversion (rather than $l = 1$) that produces shorter-period polaritons. The $l = 0 \rightarrow 1$ mode conversion mechanism was supported by the observed FT resonances at $k_0' + k_1'$ (Figures 2c-d and 3e-f). In contrast, direct “*launching high-order modes*” cannot produce such resonances.

The two studies mentioned by the reviewer involve the direct “*launching high-order modes*” without mode conversions. Our earlier study (now cited as Ref. 26) reported high-order fringes in isotopically pure hBN. These high-order modes are directly launched by the s-SNOM tip, and their standing wave interference fringes possess adequate intensity to be imaged solely due to the largely reduced loss in the monoisotopic hBN. Notably, these reduced-loss-induced high-order-mode features were unrelated to the mode conversions demonstrated in our current work. In addition, in Ref. 26, the natural hBN—also used in our current work—showed no features of high-order modes without leveraging the mode conversions reported in our current work. This distinction is directly proved in the 34nm-thick hBN in our Figure 1b: at the simple slab edge (left), no high-order features appear, consistent with Ref. 26, while at the step-shaped edge (right), clear mode-converted high-order beats are observed. The other study mentioned by the reviewer (Ref. 25) utilized the strong near-fields confined in SiC nanowires to directly launch high-order polaritons, again without reporting mode conversions. Therefore, while prior works demonstrated direct excitation of higher-order modes, neither revealed the conversion of hyperbolic polaritons between different branches—the central contribution of our current work.

To highlight the reviewer’s comment and emphasize the uniqueness of our work, we have modified the first paragraph. The corresponding sentences now read: “Two recent works^{24,25} on alpha-phase molybdenum trioxide (α -MoO₃) reveal high-order ($l > 0$) hyperbolic polaritons using geometric confinement at specific frequencies²⁴ and strong near-fields in nanowires²⁵, both of which directly launch high-order polaritons. In isotopically pure hBN²⁶, reduced phonon loss enables direct imaging of high-order polariton fringes. While these studies probe different-order hyperbolic polaritons, their distinct momenta and mode profiles make them relatively independent, lacking evident interaction, integration, or conversion.”

Comment 1-3: *It is suggested to visualize the data in Figure 4i-j graphically instead of list all values in a table.*

Response to 1-3: Following the reviewer’s suggestion, we have replaced the tables with 3D bar graphs for Figures 4i-j in the revised main text. The updated Figure 4 is copied below.

Figure 4 | Finite-difference frequency-domain (FDFD) simulation of hyperbolic polariton mode conversions in the hBN terrace. a–d, The normalized x -component of electric field (E_x) for (a) zeroth-order, (b) first-order, (c) second-order, and (d) third-order hyperbolic phonon

polaritons. **e, g**, False color maps of E_x at the cross-sections of the simple slab (symmetric, **e**) and step-shape (asymmetric, **g**) hBN terraces, with the incidence of zeroth-order hyperbolic polaritons at the black bold lines. The green boxes denote the locations of the simple slab hBN and the step-shaped hBN terraces in our simulations. In the simple slab edge (**e**), the left side is extended to infinity. In the step-shape edge (**g**), both sides are extended infinitely. **f, h**, FT spectra of the reflected fields in (**e**) and (**g**), respectively. The momenta kl of various-order hyperbolic polaritons agree well with our experimental results in Figure 2. The arrows mark the polariton modes at different branches. **i, j**, 3D bar graphs of R_{ij} : the energy ratio between the reflected j -th order mode due to the incident i -th order mode from hBN terraces featuring the simple slab and step-shape edges, respectively.

Comment 1-4: *More detail needs to be added on how the data were processed from Figure 5d to Figure 5e. Although the reviewer understood the process, a clearer explanation would make it accessible to a broader readership of Nature Communications.*

Response to 1-4: We thank the reviewer for this comment. These details were provided in Supplementary Section S3. To highlight the reviewer's suggestion, we have modified the corresponding sentence on Page 6 of the revised main text. It now reads: "This trend is further revealed by the s-SNOM line profiles (Figure 5d) and the extracted $l = 0 \rightarrow 1$ polariton line profiles from the inverse FT analysis (Figure 5e, more details about the FT analysis are provided in Supplementary Section S3)."

Comment 1-5: *A relevant work could be cited and compared (<https://doi.org/10.1021/acsnano.4c03630>).*

Response to 1-5: We thank the reviewer for mentioning this relevant work. In the revised manuscript, we have cited this paper (Ref. 34) and highlighted its merits of polariton propagation redirection in the conclusion paragraph of our revised main text. The corresponding sentence reads: "Future works may be directed toward tailoring edges or implementing metastructures at the edges for on-demand polariton mode conversion and propagation redirection³⁴ at specific orders."

Responses to Reviewer 2

In the following, we respond to the comments of this reviewer.

Comment 2-1: *The authors report mode conversion between different dispersion orders of hyperbolic polaritons achieved through a stepped van der Waals (vdW) terraces. Higher-order modes of hBN and α -MoO₃ were imaged using s-SNOM, respectively. However, I think the novelty of this proposition may not be sufficient to support its publication in Nature Communications. The excitation at edges and the observation of higher-order hyperbolic polaritons have already been extensively investigated in this field, as exemplified by the previous reports on hBN and its isotopic heterostructures, as well as the higher-order hyperbolic polaritons in α -MoO₃ launched by 3C-SiC nanowires. Thus, the authors' research represents only a tiny part of the polariton field, and its significance is relatively limited.*

Response to 2-1: We thank the reviewer for this comment. We have replied to a similar comment in Response to 1-2. Briefly, while high-order hyperbolic polaritons have been observed, branch-to-branch polariton mode conversion—the central contribution of our work—has not been studied.

Comment 2-2: *Additionally, the manuscript is not organized well, which makes it difficult for readers to understand. Figure 1 appears somewhat disorganized, with the numbering positions seeming inappropriate. Furthermore, the representation of the red dashed boxes is unclear—should an additional schematic be included to illustrate the relative positions between layers?*

Response to 2-2: We thank the reviewer for this helpful comment. In the revised main text, we have repositioned the thickness labels for the hBN terrace. In addition, following the reviewer's suggestion, we have added the optical microscope image and an additional schematic to illustrate the hBN terrace. The updated figure is copied below.

Figure 1 | Nano-infrared imaging of hyperbolic phonon polariton mode conversion in the hBN terrace using scattering-type scanning near-field optical microscopy (s-SNOM). a, The

energy-momentum (ω - k) dispersion of hyperbolic phonon polaritons in hBN (thickness: 34 nm). The red arrow indicates the mode conversion of the zeroth-order ($l = 0$) hyperbolic polariton into high-order ($l = 1, 2, \dots$) hyperbolic polaritons. **b**, s-SNOM amplitude image of the terraced hBN at the infrared (IR) frequency $\omega = 1407 \text{ cm}^{-1}$. The solid and dotted black arrows represent the incident and backscattered IR light at the s-SNOM tip (grey). The concentric orange circles delineate the hyperbolic phonon polariton waves launched by the s-SNOM tip in the terraced hBN. **c**, the optical microscope image of terraced hBN made by the pickup-and-stack technique from a mechanically exfoliated vdW thin slab. **d**, the schematic of the hBN terrace featuring a variety of simple slab, step, and covered step edges. **e**, Zoom-in s-SNOM image in **(b)**. **f**, **g**, Schematics of reflected hyperbolic phonon polaritons at simple slab edge **(f)** and step-shape edges **(g)**. Mode conversion occurs at the step-shape edges (red dashed box) in the hBN terrace **(g)**.

Comment 2-3: *In the third line of the third paragraph on page 3, the authors mention “Hexagonal boron nitride (hBN) and α -MoO₃ terraces were assembled using the tear-and-stack technique from mechanically exfoliated vdW thin slabs,” which led me to mistakenly assume that Figure 1 contained a heterostructure of hBN and α -MoO₃. However, neither Figure 1 nor Figure 2 seems to include any content related to α -MoO₃. Is this worth mentioning?*

Response to 2-3: We thank the reviewer for pointing out this potential confusion. To clarify, we have revised the sentence on Page 3 to remove α -MoO₃, which now reads: “Hexagonal boron nitride (hBN) terraces were assembled using the pickup-and-stack technique²⁷ from mechanically exfoliated vdW thin slabs (Figures 1c-d).” In addition, we have added one sentence on Page 5 to describe the similar method of assembling the α -MoO₃ terraces: “The α -MoO₃ terraces were fabricated using the similar pickup-and-stack technique²⁷ as for hBN (Figure 1).”

Comment 2-4: *In Figure 2, panels (c) and (d) refer to “step” and “covered step” structures, respectively, but there is no comparative discussion distinguishing the two. Can it be assumed that there is no difference between them?*

Response to 2-4: We thank the reviewer for raising this excellent point. In the revised manuscript, we have added the Supplementary Section S5 to compare the polariton mode conversion at the “step” and “covered step” edges of the hBN terraces in Figure 1b. With identical lower-side thicknesses and a shared higher side, the uncovered step edge exhibits a stronger $l = 0 \rightarrow 1$ fringes and therefore a larger polariton mode conversion rate than the covered step edge. We attribute this difference to the steeper step of the uncovered edge, which enhances scattering of the polariton waves compared to the covered step edge. The full discussion is provided in Supplementary Section S5 (copied below).

S5. Polariton mode conversion at the uncovered step vs. the covered step

In this section, we compare the $l = 0 \rightarrow 1$ polariton mode conversions from the two 51nm-17nm steps in the top part of the hBN terrace in Figure 1d, featuring a covered step edge (left) and an uncovered step edge (right). The two steps share identical lower-side thicknesses and the same higher side, but the uncovered one exhibits a steeper edge (Figure S6a). This steeper geometry more effectively scatters the incident polaritons, resulting in a higher $l = 0 \rightarrow 1$ mode conversion rate and stronger fringe oscillations (Figure S6b).

Figure S6 | Polariton mode conversion at the uncovered step vs. the covered step. **a**, The AFM topography across the uncovered (black) and covered (red) steps from Figure 1b in the main text. **b**, the s-SNOM line profiles of the mode converted $l = 0 \rightarrow 1$ polaritons from the uncovered (black) and covered (red) steps. Both steps share the same lower side thickness ($h = 17$ nm) and the higher side thickness ($H = 51$ nm). IR frequency $\omega = 1407$ cm^{-1} .

This excellent point raised by the reviewer inspires future work on studying the dependence of the mode conversion on the terrace edge steepness. In order to create a series of edges featuring different steepnesses while maintaining identical lower and higher side thicknesses in a controlled manner, the pickup-and-stack technique should be systematically leveraged and varied in the terrace assembly. A comprehensive s-SNOM characterization and a careful analysis of the polariton waves will then be performed to examine how the edge steepness affects the polariton mode conversion rates. Therefore, this promising study, inspired by the reviewer’s comment, deserves an independent future work. To highlight this excellent point raised by the reviewer, we have added a sentence in the conclusion paragraph: “The polariton mode conversion may be varied by delicately controlling the steepness of the asymmetric edges.”

Comment 2-5: *In Figure 4, the caption does not explain what the green solid boxes represent—should this be clarified? Additionally, in Figures (f) and (h), there appears to be a distinct peak at $l = 0 \rightarrow 2$. Why is this the case?*

Response to 2-5: We thank the reviewer for this comment. In the revised main text, we have clarified the meaning of the green solid boxes in the Figure 4 caption: “The green boxes denote the locations of the simple slab hBN and the step-shaped hBN terraces in our simulations.” In Figures 4f-h, the weak peaks at $l = 0 \rightarrow 2$ correspond to the mode conversion from the zeroth-order to the second-order mode. Although much weaker than the $l = 0 \rightarrow 1$ conversion, they are still visible in the simulation results plotted on a logarithmic scale. We have added a sentence in the figure caption to describe these peaks: “The arrows mark the polariton modes at different branches.”

Responses to Reviewer 3

We thank the reviewer for finding our results “*The data are beautiful and convincing.*” In the following, we respond to the comments of this reviewer.

Comment 3-1: *Unless I am missing something, I see a fundamental issue in the interpretation of the emergence of higher-order polariton fringes as originating from mode conversion. Let me explain my concern: Throughout the paper, the authors assume that the tip launches a fundamental $l=0$ mode, that is then converted into a $l=1$ mode upon reflection at the stepped edge. I don't quite understand how this assumption is justified. Previous work, e.g. Giles et al. Nat. Mat. 17, 134 (2018), which the authors fail to cite, has observed higher order modes without stepped edges, meaning that, in principle, higher order mode launching and detection is possible in hBN, but is usually suppressed from materials losses. What the current work successfully shows is that, in addition, the reflection of odd-order modes at simple edges is suppressed (not 0 as above work shows), but enabled again at stepped edges. I see the data in the following light: indeed, the stepped edges enable reasonably efficient reflection of odd-order hyperbolic modes, enabling their observation in a tip-launched edge-reflected scenario. However, this does not mean that the $l=1$ modes emerge from mode conversion, but simply that the odd higher-order modes launched at the tip are made visible by the stepped edge. I believe this is a reflection phenomenon rather than a conversion.*

Response to 3-1: We thank the reviewer for this thoughtful comment. The reviewer is absolutely correct that the tip-launch-edge-reflect can generate the interference fringes of the $l = 1$ mode, as demonstrated in our previous study (Ref. 26) by leveraging the largely reduced loss in monoisotopic hBN. This mechanism, however, yields polariton fringes with a periodic resonance of $\Delta = 2k_1'$ in the FT spectrum. By contrast, in our current work, the $l = 1$ fringes consistently exhibit a periodic resonance of $\Delta = k_0' + k_1'$. As detailed on Pages 4–5 and in Supplementary Section 1, and confirmed by our ω - k dispersion calculation and FDFD simulation, this specific periodic resonance ($k_0' + k_1'$) arises from the $l = 0 \rightarrow 1$ mode conversion. We therefore conclude that the higher-order mode features observed here arise from the polariton mode conversion. In the revised main text, we have emphasized the validation of our mode conversion mechanism from the ω - k dispersion calculation. The corresponding sentences on Page 5 read: “The standing wave interference between the converted first-order and the newly launched zeroth-order polaritons produces a periodic resonance of $\Delta = k_0' + k_1'$ (See details in Supplementary Section 1). Therefore, the momentum k_1' of the converted $l = 0 \rightarrow 1$ hyperbolic polaritons can be extracted from the FT spectra in Figures 2c-d. These extracted s-SNOM data (red and blue dots) from both the step-shape edge and the covered step-shape edge reveal a systematic ω dependence and agree well with our ω - k dispersion calculation (false color maps) in Figures 2f-g, thereby validating the polariton mode conversion mechanism described above.”

Comment 3-2: *The problem of interpretation/assumption is particularly apparent in figure 4 where the authors explicitly assume an $l = 0$ fundamental mode as source, excluding the $l = 1$ component contained in the tip-launched hyperbolic mode. Without a scale on the log-plots (f, h), it is also difficult to gauge how efficient the mode conversion process is. Furthermore, it is very difficult to grasp in Fig. 4 where the slab edge (simple or stepped) are in these plots.*

Response to 3-2: We thank the reviewer for this comment. In Figure 4, we have considered the tip-launched hyperbolic polaritons for all branches ($l = 0, 1, 2, 3$, etc.), not only for $l = 0$. These results are provided in Figures 4i-j. As described on Page 5, the symmetric simple slab edge

preserves polaritons in the same order (same l , see details in Figure 4i). In contrast, the asymmetric step-shape edge does not preserve l but leads to strong mode conversion (Figure 4j). The detailed discussion was provided on Pages 5–6 in the main text: “In Figures 4i-j, we quantify the mode conversion rate using reflectivity R_{ij} —the energy ratio between the reflected j -th order mode due to the incident i -th order mode—for more generalized cases where the incident hyperbolic polaritons are at $l = 0, 1, 2$, and 3. Notably, the symmetric simple slab edge preserves the polariton modal symmetry (Figure 4i): R_{00} , R_{11} , R_{22} , and R_{33} are significantly larger than others, indicating that even (odd) modes $l = 0, 2, \dots$ ($1, 3, \dots$) are only reflected into even (odd) modes. Conversely, the asymmetric step-shape edge reflects polaritons into both odd and even modes, regardless of the order of the incident polaritons (Figure 4j)—demonstrating that breaking edge symmetry leads to polariton mode conversion.” We agree with the reviewer that the tip does launch $l = 1, 2, 3$, etc. polaritons. However, due to their much shorter propagation length, these high-order polaritons cannot form evident standing wave fringes in the standard tip-launch-edge-reflect mechanism unless the loss is largely reduced using monoisotopic crystals, as detailed in our Response to 1-2. On Page 4–5 of the main text, we have the corresponding comment: “Notably, high-order hyperbolic polaritons exhibit much shorter propagation lengths than their zeroth-order counterpart. Therefore, without the mode conversion at the simple slab edge, it is difficult for high-order polaritons to complete a tip-edge roundtrip to produce evident standing wave interferences (Figure 1b and ref. ^{11, 15-23}).”

To address the reviewer’s other points, we have made the following revisions. We have added vertical scales to Figures 4f-h, where the $l = 0 \rightarrow 1$ mode conversion exhibits considerable efficiency. In addition, we have boldened the contour lines representing the simple slab and stepped edges in the right panels of Figures 4e and 4g to highlight their locations in our simulations. The updated Figure 4 is shown in our Response to 1-3.

Comment 3-3: *I am struggling with the momentum conservation concept for the mode conversion. Since the momentum of the $l=1$ mode is not a harmonic multiple (typically) of the $l = 0$ mode, I am not actually sure how a coherent mode conversion would work. For a harmonic (in frequency domain commonly known as high harmonic generation), it would be more obvious, but I would expect that a non-integer harmonic would in fact largely cancel out. So I don’t quite understand this conceptually, and would urge the authors to elaborate.*

Response to 3-3: We thank the reviewer for commenting on the mode conversion concept. We agree that the $l = 1$ mode is not a harmonic multiple of the $l = 0$ mode, and the mode conversion described here is therefore distinct from harmonic generation in the frequency domain. In our case, the stepped edge breaks the structure symmetry and relaxes the strict momentum conservation requirement, allowing scattering between polariton modes with different l . On Page 4 of the main text, we have elaborated this mode conversion process: “In contrast, a step-shape edge (Figure 1g) introduces a sharp step corner that can strongly scatter^{22, 30} the incoming polaritons inside the slab. This scattering provides additional momentum^{22, 30}, enabling mode conversion by bridging the k -mismatch between polaritons of different orders. As a result, zeroth-order ($l = 0$) polaritons are converted into first-order ($l = 1$) modes upon reflection at the step-shaped edge.” The coherence of this process is revealed experimentally by the interference fringes and in the FT spectra, where the characteristic periodic resonance $\Delta = k_0' + k_1'$ uniquely signals the interference between the $l = 0$ and $l = 0 \rightarrow 1$ polaritons. As detailed on Pages 5–6, this observation is supported by our ω - k dispersion calculations and FDFD simulations, thereby confirming that symmetry breaking at the stepped edge enables coherent mode conversion between non-harmonic polariton branches.

Comment 3-4: *In the conclusion, the authors claim not just mode conversion but also energy redistribution. This is a bit odd since all modes are at the same energy.*

Response to 3-4: We thank the reviewer for the comment. We agree that all polariton modes studied here lie at the same excitation energy (ω), and thus, there is no redistribution among different photon energies. Our intended meaning was that the incident polariton energy flux is redistributed among different polariton orders (e.g., from $l = 0$ into $l = 1, 2$, etc.) upon scattering at asymmetric step edges. To avoid potential confusion, we have removed the term “energy redistribution” from our revised manuscript.

Response to referees' comments

Manuscript #: NCOMMS-25-54301A

Responses to Reviewer 1

We thank the reviewer for finding that our revision “*addressed most of the reviewers' comments.*” In the following, we respond to the comments of this reviewer.

Comment 1-1: *The new Figure 1 is still busy and crowded. Maybe it could be split into two figures.*

Response to 1-1: We thank the reviewer for this constructive suggestion. In the revised main text, we have reorganized and simplified Figure 1 (copied below).

Figure 1 | Nano-infrared imaging of hyperbolic phonon polariton mode conversion in the hBN terrace using scattering-type scanning near-field optical microscopy (s-SNOM). a, The energy-momentum (ω - k) dispersion of hyperbolic phonon polaritons in hBN (thickness: 34 nm). The red arrow indicates the mode conversion of the zeroth-order ($l = 0$) hyperbolic polariton into

high-order ($l = 1, 2, \dots$) hyperbolic polaritons. **b**, The optical microscope image of terraced hBN made by the pickup-and-stack technique from a mechanically exfoliated hBN thin slab. **c**, The schematic of the hBN terrace featuring a variety of simple slab, step, and covered step edges. **d**, s-SNOM amplitude image of the terraced hBN at the infrared (IR) frequency $\omega = 1407 \text{ cm}^{-1}$. The solid and dotted black arrows represent the incident and backscattered IR light at the s-SNOM tip (grey). The concentric orange circles delineate the hyperbolic phonon polariton waves launched by the s-SNOM tip in the terraced hBN. **e**, Schematics of mode conversion and no mode conversion of hyperbolic phonon polaritons at the step-shape edge (top) and the simple slab edge (bottom), respectively. **f**, Cross-section schematics of the step-shape edges (top) and simple slab edge (bottom) in (**d**).

Comment 1-2: *The reported absolute conversion efficiency in SI is very low (below 8%). Can the authors comment on this? Also, I still think the absolute value should be reported in the main text instead of the normalized value currently shown there.*

Response to 1-2: We thank the reviewer for this helpful suggestion. Following the recommendation, we have updated Figure 5f to display the absolute values of the mode conversion rate. At the infrared (IR) frequency $\omega = 1407 \text{ cm}^{-1}$, the absolute mode conversion rate $R_{01} = 8\%$ (for the terrace with thicker side thickness $H = 51 \text{ nm}$ and thinner side thickness $h = 17 \text{ nm}$) is actually quite considerable for momentum (k)-change polariton processes. For context, the most common k -change polariton process—the polariton tip-launching under the IR illumination—has an efficiency of less than 3% (defined as the power ratio between launched polaritons and the incident IR beam) when a focused $\sim 10 \text{ }\mu\text{m}$ spot illuminates a $\sim 10 \text{ nm}$ tip. In our system, during the $l = 0 \rightarrow 1$ mode conversion at the step-shape edge, 55% of the incident $l = 0$ polariton power is transmitted into the thinner side of the terrace, and 14% is scattered into the free space. Therefore, only 31% of the original polariton power was reflected: 23% remains in the $l = 0$ mode and 8% converts into the $l = 1$ mode—the $l = 0 \rightarrow 1$ mode conversion. Therefore, the 8% mode conversion rate R_{01} is a meaningful value, particularly considering the competing transmission and scattering (into the free-space) channels. We also note that this conversion rate can be optimized by varying the ω , H , and h to reduce the transmission and free-space scattering at the step-shape edge. Additionally, anti-transmission structures could be implemented on the thinner side to further suppress the transmission and thus enhance the mode conversion rate.

Comment 1-3: *Although the authors confirm that the higher-order mode arises from mode conversion, could a boundary-induced excitation of the higher-order mode also play a role here, or could such a mechanism explain the observed phenomenon, as demonstrated in a recent work (<https://doi.org/10.1038/s41566-025-01755-5>)?*

Response to 1-3: We thank the reviewer for mentioning this related work. In the revised main text, we have highlighted this work: “During the manuscript peer review process, we became aware of a related study reporting boundary-exited high-order hyperbolic phonon polaritons and the pseudo-birefringence effect in $\alpha\text{-MoO}_3$ ³⁷.” This related work reports a similar mode conversion mechanism: The initially launched $l = 0$ polaritons were converted to the $l = 1$ polaritons by the strong scattering of the underneath Au boundary—the so-called “*boundary-induced excitation of the higher-order mode*.” The Au boundary in this related work plays a similar role to the step-shape edge in our work for the polariton mode conversions.

Responses to Reviewer 2

Comment 2-1: *The claimed novelty of mode conversion is unconvincing and insufficiently supported by evidence. The central claim of the mode conversion mechanism at asymmetric step edges is not adequately distinguished from prior reports in this field. The manuscript fails to articulate a clear conceptual advance beyond the established framework. Crucially, the experimental data and analysis presented are insufficient to support this claim robustly. Furthermore, the treatment of high-order polaritons remains superficial, focusing on their appearance rather than a deep exploration of their converted modal properties, efficiency, or further tuning. The authors are encouraged to refer to recent in-depth studies on higher-order polaritons (DOI: 10.1002/adma.202300301; DOI:10.1038/s41566-025-01755-5) to see the level of mechanistic insight and evidence required for a high-impact claim.*

Response to 2-1: We thank the reviewer for this comment. We emphasize that our study uniquely demonstrates polariton mode conversion, which, to our best knowledge, has not been previously reported. This mode conversion mechanism is supported by both experimental data and numerical simulations, consistent across the real-space images and energy–momentum (ω - k) dispersions. We have demonstrated polariton mode conversion in two representative van der Waals polaritonic crystals (hBN and α -MoO₃) and suggested that it is generic across polaritonic systems. Furthermore, we have systematically examined mode conversion between multiple polariton orders, quantified the conversion efficiency, and revealed its tunability by varying the terrace height ratio. We believe these results clearly articulate the novelty and mechanistic insight of our work. In the revision, we have also thoroughly proofread and polished the manuscript for improved clarity, and we have highlighted the two works mentioned by the reviewer. We would, of course, be happy to address any further specific comments from the reviewer or the editor.

Comment 2-2: *Poor organization and confusing figure presentation undermine the clarity. The figures, which are critical for understanding the nano-optical phenomena described, are poorly organized and inconsistently labeled. High-quality visual presentation is a fundamental aspect of scientific communication, and the current standard falls short of what is expected for a journal of this caliber.*

Response to 2-2: We thank the reviewer for the feedback. In both the previous and current revisions, we have incorporated all figure-related suggestions from reviewers. In addition, we have carefully refined all figures to ensure clear organization, consistent labeling, and high visual quality. Specifically, in the current revision, we have:

- a. Reorganized and simplified Figure 1 for improved clarity;
- b. Updated the false-color plots for the simulated ω - k dispersions in Figures 1–3 by removing the background color to enhance the visibility of the polariton branches;
- c. Extended the Fourier Transform (FT) spectra in Figures 2–4 for a broader data range; and
- d. Updated the ω - k dispersion unit to μm^{-1} for consistency with the FT spectra.

We respectfully note that the present comment does not specify which figures remain unclear. We therefore believe the revised figures now meet the publication standards of Nature Communications. Again, we would, of course, be happy to further address any concrete, figure-specific suggestions from the reviewer or the editor.

Responses to Reviewer 3

We thank the reviewer for finding our revision “*responded well to my previous review as well as the other reviews,*” and “*Conceptually, I see now how one would distinguish the edge-converted from the directly launched higher order modes.*” In the following, we respond to the comments of this reviewer.

Comment 3-1: *However, apart from the mathematical explanation in section S1, I do not see this proven in any of the data. What I would need to see is where in the FT spectrum to expect the direct (tip-launched edge-reflected) higher-order modes vs. edge-converted. It is in fact quite misleading, e.g., in Fig. 2 to show the dispersion (where all this treatment/extraction was already done) is plotted right next to the experimental data on different scales, absolute and relative, respectively, so by eye I would think (and was thinking that in the first round of reviews) that the data and the dispersion scales are the same (apart from a factor of 2). The $l=0$ momentum is so small, that the ratio between $l=0$ and $l=1$ momenta is qualitatively similar to the FT peak positions for $l=0$ and $l=0 \rightarrow 1$. If I understand correctly (now), the $l=1$ direct launching peak would emerge at almost twice the momentum as the $l=0 \rightarrow 1$ peak? This is a critical point that needs to be made abundantly clear, and, honestly, it is not clear at all, not in any of the main text figures, nor in the SI. Part of the issue still is the simulations explicitly only exciting the $l=0$ mode. Or, like in Fig. S3, this critical comparison is not done. If there, I would compare the $l=1$ peak position with $l=1$ source, and the $l=0 \rightarrow 1$ peak position with $l=0$ source, it would be clear - bam. But the author never make this comparison visually or in numbers. In fact, it is worse: in Fig 3 e,f and probably also in Fig. 2 c,d, the range of the FT spectra are cropped such that the any directly imaged $l=1$ reflected mode would not even be in the range of the plot. So, the data proving this mechanism are not actually shown!*

Response to 3-1: We thank the reviewer for this insightful comment. We clarify that the experimental data points in Figures 2e-g and 3g-h were extracted from the $l=0 \rightarrow 1$ mode conversion. Specifically, the momentum k_1' was obtained using the relation $\Delta = k_0' + k_1'$, following the interference mechanism of the mode conversion described in Supplementary Section S1. The values of Δ and k_0' were directly read from the resonance positions in the FT spectra (Figures 2b-d and 3e-f) of our s-SNOM data. Following the mode conversion induced interference mechanism, these extracted data (dots) agree well with our calculated ω - k dispersions (false color maps) for both hBN (Figures 2e-g) and α -MoO₃ (Figures 3g-h), thereby supporting the polariton mode conversion phenomenon reported in this work.

To address the reviewer’s comment about the plot unit and range, we have revised the figures accordingly. We now use a consistent momentum unit of μm^{-1} in all FT spectra and ω - k dispersions. We have also extended the horizontal Δ axis in the FT spectra to display a broader range of data. The reviewer is correct that there may be weak signals around $\Delta \sim 65 \mu\text{m}^{-1}$ in Figures 2c-d and 3e-f. While these values are close to the expected positions of the tip-launch-edge-reflect $l=1$ polariton fringes, these FT signals are more than an order of magnitude weaker than the $l=0 \rightarrow 1$ mode conversion resonances and thus cannot account for observable real-space signatures in our s-SNOM data.

To highlight the reviewer’s comment, we have made the following revisions in the text:

- a. We have commented on the tip-launch-edge-reflect interference mechanism and the weak FT signals at high μm^{-1} in the main text: “Notably, high-order hyperbolic polaritons exhibit much shorter propagation lengths than their zeroth-order counterpart. Therefore, without the mode conversion, it is difficult for high-order polaritons to complete a tip-launch-edge-reflect round trip to produce evident standing wave interferences at either simple slab edges

(Figure 1d black box and ref. ^{11, 15-23}) or step-shape edges. Note that weak signals around $65 \mu\text{m}^{-1}$ cannot correlate with observable s-SNOM signatures, especially the $l = 0 \rightarrow 1$ beats (see the FT decomposition of our s-SNOM data in Supplementary Section S2).”

- b. We have added the Supplementary Section S2 to show the decomposition of the s-SNOM data through the FT analysis (copied below). We demonstrated that the main features of our s-SNOM data—the small-period beats—arise from the $l = 0 \rightarrow 1$ mode conversion, whereas the potential tip-launch-edge-reflect $l = 1$ polariton fringes are negligible.

S2. Decomposition of the s-SNOM data via FT analysis

This section provides the decomposition of a representative s-SNOM line profile (Figure S2a, reproduced from Figure 2c in the main text) into its constituent real-space oscillation components based on the corresponding FT resonances (Figure S2a). In Figure S2b, we perform inverse FT transforms using the filtered resonances marked in red, blue, and pink in Figure S2a. They correspond, respectively, to standing wave interferences for edge-launch-photon-interfere + tip-launch-edge-reflect $l = 0$ polaritons (red), $l = 0 \rightarrow 1$ converted polaritons (blue), and tip-launch-edge-reflect $l = 1$ polaritons (pink). In comparison to the s-SNOM line profile (black curve in Figure S2b), it is evident that the s-SNOM long-period fringes arise from the $l = 0$ polaritons, while the short-period beats originate from the $l = 0 \rightarrow 1$ converted polaritons (blue). In contrast, the tip-launch-edge-reflect $l = 1$ polaritons correspond to the pink resonance that is orders of magnitude weaker than the red and blue ones (Figure S2a) and their real-space features (pink curve, Figure S2b) cannot be observed from the s-SNOM line profile (black). This conclusion is further confirmed in Figure S2c: the orange curve, obtained by summing only the $l = 0$ and $l = 0 \rightarrow 1$ components, already reproduces all observable features of the experimental data. Adding the tip-launch-edge-reflect $l = 1$ component (green curve) does not introduce any additional visible features, confirming its negligible contribution.

Figure S2 | The decomposition of s-SNOM line profile by FT analysis. **a**, The FT spectrum of the s-SNOM line profile (reproduced from Figure 2c). Red, blue and pink rectangles mark the resonant features. **b**, The inverse FT transforms of the filtered regions in **(a)**. The black curve show the s-SNOM line profile (reproduced from the red curve in Figure 2a). **c**, Green: the sum of red, blue, and pink components in **(b)**. Orange: the sum of red and blue components in **(b)**. Black: the s-SNOM line profile. Frequency: 1407 cm^{-1} .

Comment 3-2: *One additional comment: a much cleaner way of demonstrating mode conversion was just published in Nature Photonics (<https://doi.org/10.1038/s41566-025-01755-5>), and the authors must at least mention this work (The authors became aware of a competing work that appeared during preparation of this manuscript ... or something like that). Importantly, this other*

work uses nano-antennas instead of the tip to launch the polaritons, making much clearer images and a much cleaner analysis. Additionally, they also explicitly proof that the momentum parallel to the edge is conserved during the conversion, as to my previous question about momentum conservation.

Response to 3-2: We thank the reviewer for mentioning this useful reference. Following the reviewer's suggestion, we have highlighted this important work at the end of our main text: "During the manuscript peer review process, we became aware of a related study reporting boundary-exited high-order hyperbolic phonon polaritons and the pseudo-birefringence effect in α -MoO₃³⁷."

Response to referees' comments

Manuscript #: NCOMMS-25-54301B

Responses to Reviewer 3

Comment 3-1: *Noh and coworkers have improved some of the aspects that I (and the other reviewers) had raised. I am still not particularly happy with the way the story is presented, brushing over the key observations that distinguish the current work from many others in the field. For instance, Eq. 1, has been shown in many previous papers. Instead, I would appreciate an Eq. that specifically discussed the mode conversion. However, I feel that with minimal changes during each iteration of the review process, this is not going to change much.*

I do have some concerns, however, specifically relating to the changes included in the last iteration. As per my request, the authors now show the full momentum range including the $l=1$ tip-launched edge reflected resonance in Fig. 2c,d that I had wondered about. In the SI, they also discuss this feature as " $l=1$ tip". However, in the revised main text, the authors state "Note that weak signals around $65 \mu\text{m}^{-1}$ cannot correlate with observable s-SNOM signatures", and support this statement with SI Fig. S2c. I find this whole argument (I cannot see the difference by eye if I remove the respective Fourier component) highly problematic and - in fact wrong. The Fourier component for the $l=1$ tip contribution is clearly observed in the data, and only because I don't "see it" in the real space traces doesn't make it less part of the data. Even more so, in the main text and Fig. 2c,d, the authors do not even call this by its name, apparently trying to hide the fact that there is a $l=1$ tip signal. This is borderline in terms of good scientific practice. I strongly advise against such a practice. More importantly, I don't understand why the authors do this. There is enough prior work that shows that higher order modes can indeed be observed also without mode conversion, as for instance pointed out by reviewer 2. From this current work, it is clear that the efficiency of mode conversion to $l=1$ is much higher than the direct excitation/edge reflection. To be more quantitative, the authors could in fact use these data to prove the claim, since the mode-converted $l=1$ contribution should have a different decay length than the $l=1$ tip contribution.

Response to 3-1: We thank the reviewer for the thoughtful and detailed feedback. We agree that Eq. 1 has appeared in many previous works. However, without explicitly presenting this equation in the main text, it would be difficult for us to clearly define and discuss polaritons of different orders throughout the manuscript. For this reason, we believe it is beneficial to retain Eq. 1 while citing the relevant literature. We appreciate the reviewer's suggestion to include an equation specifically describing the mode-conversion process. Developing such an analytical expression would require a systematic theoretical treatment of mode conversion, which we view as an important but independent direction for future work.

Following the reviewer's advice, we have now explicitly labeled the weak features in Figures 2c and 2d as " $l = 1$," and we have revised the corresponding text to avoid any ambiguity regarding these features. In particular, we replaced the term "observable" with "dominant" to clarify our intention. The revised sentence reads: "Note that weak signals around $65 \mu\text{m}^{-1}$ —which by their value may correspond to the tip-launched–edge-reflected $l = 1$ polariton fringes—cannot correlate with the dominant s-SNOM signatures, especially the $l = 0 \rightarrow 1$ beats (see the FT decomposition of our s-SNOM data in Supplementary Note 2)."

We also appreciate the reviewer's suggestion to analyze the data using the decay lengths of the different fringes. We fully agree that the decay lengths of the tip-launched–edge-converted

($l = 0 \rightarrow 1$) and tip-launched–edge-reflected ($l = 1$) polariton fringes differ, consistent with their distinct oscillation periods shown in Figures 2c,d. However, we respectfully maintain that an extended discussion of the weak $l = 1$ tip-launched–edge-reflected features would be redundant and would detract from the central focus of our work. These weak features are not directly related to our primary finding—the clear small-period beats associated with the $l = 0 \rightarrow 1$ Fourier resonances that emerge exclusively at step-shaped edges. The mode-conversion signatures remain the dominant and unambiguous features in both the real-space and Fourier-transformed data, and our interpretation does not rely on the weak $l = 1$ reflections, which have been established in prior literature.